# Evolution of Cooperation for Multiple Mutant Configurations on All Regular Graphs with $N \leq 14$ Players

**Hendrik Richter**[ID]

Faculty of Engineering, HTWK Leipzig University of Applied Sciences, D-04251 Leipzig, Germany;
hendrik.richter@htwk-leipzig.de; Tel.: +49-341-3076-1123

**Abstract:** We study the emergence of cooperation in structured populations with any arrangement of cooperators and defectors on the evolutionary graph. In a computational approach using structure coefficients defined for configurations describing such arrangements of any number of mutants, we provide results for weak selection to favor cooperation over defection on any regular graph with $N \leq 14$ vertices. Furthermore, the properties of graphs that particularly promote cooperation are analyzed. It is shown that the number of graph cycles of a certain length is a good predictor for the values of the structure coefficient, and thus a tendency to favor cooperation. Another property of particularly cooperation-promoting regular graphs with a low degree is that they are structured to have blocks with clusters of mutants that are connected by cut vertices and/or hinge vertices.

**Keywords:** evolutionary game theory; fixation properties; structure coefficients; regular graphs; graph–theoretical properties; graph cycles

## 1. Introduction

Describing conditions for the emergence of cooperation in structured populations is a fundamental problem in evolutionary game theory [1–5]. In structured populations, the network describing which players interact with each other may be crucial for the fixation of a strategy. Recently, several attempts have been made to explore the universe of interaction graphs in order to link graph properties to fixation. For a single cooperator, this question has been studied intensively and recently relationships have been mapped for a large variety of different interaction graphs connecting which strategy is favored with the fixation probabilities and the fixation times [6–9]. These results clarify for a single mutant the relationships between the graph structure, on the one hand, and fixation probability and fixation time, on the other. The main findings are that generally fixation probability and fixation time is correlated such that a higher fixation probability comes with a higher fixation time. Within this general rule, it has further been shown that generalized stars maximize fixation probability while minimizing fixation time, while comet-kites minimize fixation probability while maximizing fixation time [7]. Furthermore, if we allow self-loops and weighted links, we may construct arbitrarily strong amplifiers of selection [8]. Compared with these findings, the problem of multiple cooperators (or more than one mutant) is far less studied. One approach uses configurations [10,11] and structure coefficients [12] and has shown that cooperation is favored over defection under conditions that can be linked to spectral graphs measures and cooperator path length [13,14].

This paper gives a computational study and uses structure coefficients defined for configurations describing any arrangement of any number of mutants. It consequently deals with strategy selection for multiple mutants on evolutionary graphs and addresses two central questions. The first is to find out which interaction network modeled as a regular graph yields the largest structure coefficient and

therefore is most suited to promote the evolution of cooperation. This is reported for all regular graphs with $N \leq 14$ vertices (=players). This question is studied subject to three parameters, the number of players, coplayers, and cooperators. Answering this question may inform the design of interaction networks with prescribed abilities to promote or suppress cooperation. As there are some trends over varying these three parameters, it appears possible to conjecture for beyond the considered parameters. The second question studied takes up the observation that there are differences in the values of the structure coefficients over regular interaction graphs and asks what makes some graphs different from others in terms of promoting the evolution of cooperation. Our main interest is what these differences are from a graph–theoretical point of view. This goes along with identifying certain properties of regular cooperation-promoting graphs. The main result is that the number of graph cycles of a certain length is a good predictor of a large value of the structure coefficient. Especially for a smaller number of coplayer, graphs that particularly promote cooperation have cycles with a small length. Furthermore, these graphs are structured to have blocks that are connected by cut vertices and/or hinge vertices. Cooperators cluster on these blocks and serve as a mutant family that may invade the remaining graph. The study presented here uses structure coefficients, which have been derived for birth–death and death–birth processes [12]. However, as the structure coefficients solely depend on the distribution of cooperators and defectors on the evolutionary graph, they could, at least in principle, also be calculated for other strategy updating processes as long as these processes are not completely random. Thus, the methodology reported here may also be applicable for other types of evolutionary dynamics, for instance, non-imitative dynamics.

The paper is structured as follows. In Section 2, the main results are given. In particular, upper and lower bounds on the structure coefficients are presented for DB updating and all interaction networks modeled as regular graphs with $N \leq 14$ players. Furthermore, it is shown that between maximal structure coefficients (and thus conditions favoring the prevalence of cooperation) and the count of cycles with a certain length, there is an approximately linear relationship. The results are discussed in Section 3, while the Appendices review the methodological framework of configurations, regular graphs, and structure coefficients, discuss graph isomorphism, and give a collection of graphs with maximal structure coefficients.

## 2. Evolution of Cooperation

### 2.1. Upper and Lower Bounds on the Structure Coefficients

The structure coefficient $\sigma(\pi, \mathcal{G})$ introduced by Chen et al. [12] (see [13,14] for further analysis) is a measure of whether or not cooperation is favored over defection in games with any arrangement of cooperators and defectors on regular evolutionary graphs. More strictly speaking, in an evolutionary game with weak selection and a payoff matrix (A1), the fixation probability of cooperation is larger than the fixation probability of defection if

$$\sigma(\pi, \mathcal{G})(a - d) > (c - b), \tag{1}$$

see also Appendix A. This condition connects the values of the payoff matrix, the structure of the evolutionary graph $\mathcal{G}$ and the arrangement of cooperators and defectors on this graph expressed by the configuration $\pi$ with long-term prevalence of cooperation. The structure coefficient $\sigma(\pi, \mathcal{G})$ generalizes the structure coefficient $\sigma$ introduced by Tarnita et al. [15] which yields the same condition for favoring cooperation, $\sigma(a - d) > (c - b)$, but applies to a single cooperator (or a single mutant). By contrast, $\sigma(\pi, \mathcal{G})$ is valid for any arrangement of cooperators and defectors on the evolutionary graph and specifically for several cooperators (or multiple mutants).

As the structure coefficient varies over configurations $\pi$ and graphs $\mathcal{G}$, it is natural to ask about upper and lower bounds of $\sigma(\pi, \mathcal{G})$. In this paper, we approach this question by checking all $\sigma(\pi, \mathcal{G})$, which appears feasible for a small number of players $N \leq 14$ and all regular graphs with up to 14 vertices. We classify the structure coefficients and graphs with respect to the number of players $N$.

Furthermore, the configurations $\pi$ are also grouped according to the number of cooperators $c(\pi)$, $2 \leq c(\pi) \leq N - 2$, while the graphs $\mathcal{G}$ are sorted according to the number of coplayers $k$ (which equals the degree of the graph). As the structure coefficients $\sigma(\pi, \mathcal{G})$ vary over configurations *and* graphs $\mathcal{G}$, we may define two bounds. A first bound is over all $2^N - 2$ non-absorbing configurations, which is called $\sigma_{max_i}$. Thus, we obtain for each graph $\mathcal{G}_i$, $i = 1, 2, \ldots \mathcal{L}_k(N)$, the quantity $\sigma_{max_i} = \max_{\pi} \sigma(\pi, \mathcal{G}_i)$. A second bound, called $\sigma_{max}$, is derived from the first bound and additionally collects over all $\mathcal{L}_k(N)$ regular graphs with a given $N$ and $k$ according to Table A1. Thus, we get $\sigma_{max} = \max_i \sigma_{max_i}$. For the minimum, the bounds are defined like-wise.

Figure 1 shows the maximal structure coefficient $\sigma_{max}$ and the maximal difference $\Delta\sigma = \sigma_{max} - \sigma_{min}$ over players $N$ and coplayers $k$ for DB updating, see Equation (A3). As discussed in Appendix B these results apply to any instance of a regular graph, for example to random regular graphs. It can be seen that the maximal structure coefficient $\sigma_{max}$ is the largest for $k = 3$, which is cubic graphs. For $k > 3$, the values of $\sigma_{max}$ get gradually smaller. In other words, the more coplayers there are, the smaller is $\sigma_{max}$. For a constant number of coplayers, $\sigma_{max}$ increases with $N$, which is the number of players. The increase, however, gets gradually smaller and converges for $N \to \infty$ to a constant, which is $\sigma(\pi, \mathcal{G}) \to \sigma = (k + 1)/(k - 1)$ [12,16]. For instance, for $k = 3$, the structure coefficients converge to $\sigma(\pi, \mathcal{G}) \to \sigma = 2$.

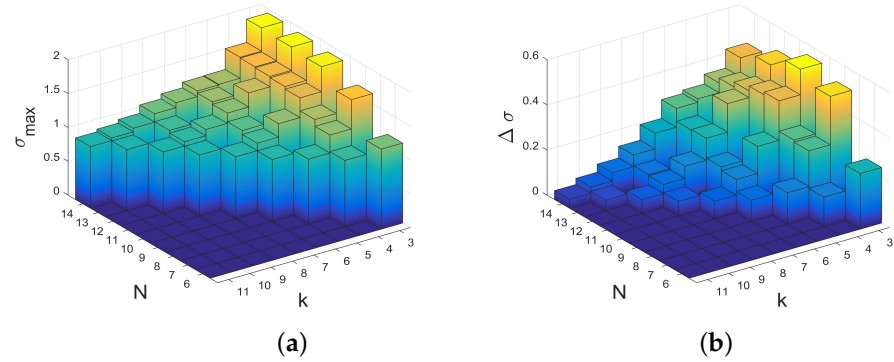

**Figure 1.** The maximal structure coefficient $\sigma_{max}$ (**a**) and the maximal difference $\Delta\sigma = \sigma_{max} - \sigma_{min}$ (**b**) over the number of players $N$ and coplayers $k$ for all regular interaction graphs with $6 \leq N \leq 14$ and $3 \leq k \leq N - 3$ according to Table A1.

In other words, for the thermodynamic limit with an infinite population, the prevalence of cooperation only depends on the number of coplayers $k$ of a regular graph, but not on the graph structure or the number and arrangement of cooperators on the graph. The largest difference between maximal and minimal structure coefficient $\Delta\sigma = \sigma_{max} - \sigma_{min}$ we also get for $k = 3$. Here, $\Delta\sigma$ increase to a largest values (for instance for $k = 3$ this happens for $N = 10$) before falling for $N$ getting even larger, converging to $\Delta\sigma = 0$ for $N \to \infty$.

We next analyze the maximal structure coefficients depending on the number of cooperators $c(\pi)$. Thus, the maximum is over all $\#_{c(\pi)} = \binom{N}{c(\pi)}$ configurations with the same number of cooperators $2 \leq c(\pi) \leq N - 2$ and all regular graphs according to Table A1. The maximal values of $\sigma_{max}$ and $\Delta\sigma$ are obtained for $c(\pi) = N/2$ for $N$ even and for both $(N + 1)/2$ and $(N - 1)/2$ for $N$ odd. An exception is $N = 12$ and $k = 3$, where $\sigma_{max}$ is obtained for $c(\pi) = 5$ and $c(\pi) = 7$. Furthermore, we get the following results, see Figure 2 as examples for $N = 12$ and $N = 14$. The value $\sigma_{max}$ and $\Delta\sigma$ are symmetric with the number of cooperators $c(\pi)$ and generally higher for the number of cooperators and defectors exactly or approximately the same as for a small number of cooperators or a small number of defectors. For the number of coplayers $k$ getting larger, the differences over the number of cooperators $c(\pi)$ for both $\sigma_{max}$ and $\Delta\sigma$ are leveled.

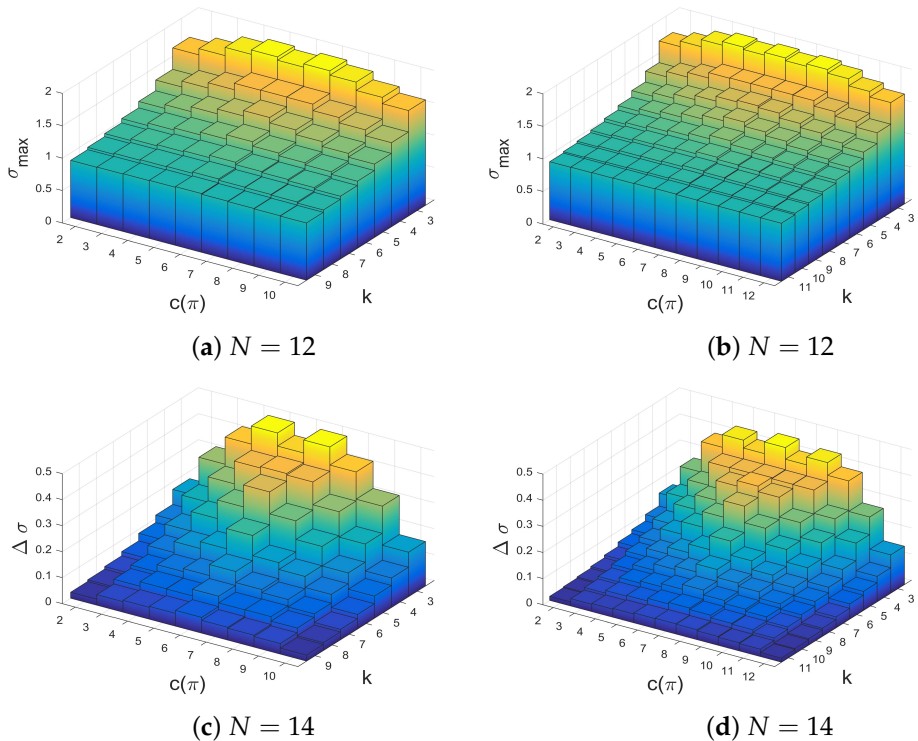

**(a)** $N = 12$

**(b)** $N = 12$

**(c)** $N = 14$

**(d)** $N = 14$

**Figure 2.** The maximal structure coefficient $\sigma_{max}$ and the maximal difference $\Delta\sigma = \sigma_{max} - \sigma_{min}$ over the number of coplayers $k$ and cooperators $c(\pi)$ for all regular interaction graphs with $N = 12$ and $N = 14$ according to Table A1.

Apart from the numerical values of the maximal structure coefficients $\sigma_{max}$ and their relations to the number of players $N$, coplayers $k$ and cooperators $c(\pi)$, it is also interesting to know for which of the $\mathcal{L}_k(N)$ graphs the maximal values occurs. We call the graphs for which this happens the $\sigma_{max}$-graphs. Their number is $\#_{\sigma_{max}}$. Table 1 gives the number of $\sigma_{max}$-graphs, $\#_{\sigma_{max}}$, for all $N$ and $k$ considered here, see also Appendix C for some examples of $\sigma_{max}$–graphs. If we compare these numbers with the total number $\mathcal{L}_k(N)$ of $k$-regular graphs on $N$ vertices, see Table A1, we observe that $\mathcal{L}_k(N)$ grows much faster than $\#_{\sigma_{max}}$. In other words, the $\sigma_{max}$-graphs become rare as $N$ increases. Figure 3 shows the quantity

$$\#_{log} = -\frac{1}{N^2} \log\left(\frac{\#_{\sigma_{max}}}{(4k - 1/4k^2)\mathcal{L}_k(N)}\right) \tag{2}$$

over $N$ and $k$ (Figure 3a), and over $c(\pi)$ and $k$ for $N = 14$ (Figure 3b). We may conclude that as a rough approximation the ratio $\frac{\#_{\sigma_{max}}}{\mathcal{L}_k(N)}$ falls exponentially in $N$ and polynomially in $k$ for $k \approx N/2$ and $N$ getting larger. Furthermore, observe from Figure 3b that for small and large values of the number of cooperators $c(\pi)$ there is a larger number of graphs that are $\sigma_{max}$-graphs. The $\sigma_{max}$-graphs become rarer for $c(\pi) \approx N/2$, for which $\sigma_{max}$ is largest as well.

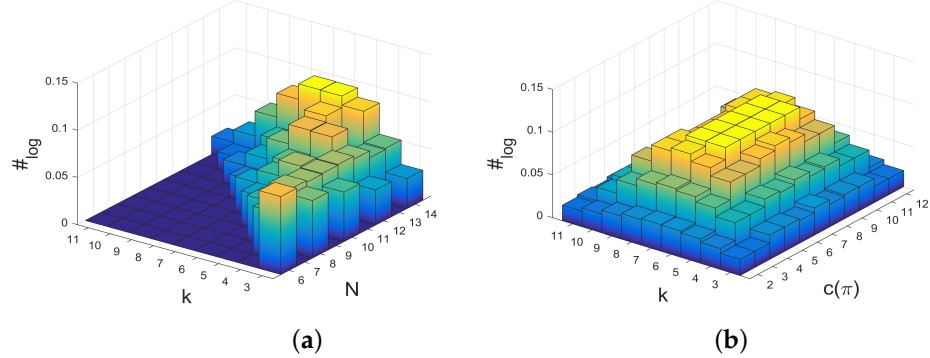

**Figure 3.** The quantity $\#_{log}$ according to Equation (2) relating the number of $\sigma_{max}$-graphs, $\#_{\sigma_{max}}$, to the number of regular graphs $\mathcal{L}_k(N)$ for players $N$, coplayers $k$, and cooperators $c(\pi)$: (**a**) over $N$ and $k$ and (**b**) over $k$ and $c(\pi)$ for $N = 14$.

**Table 1.** The numbers $\#_{\sigma_{max}}$ of graphs with maximal $\sigma_{max}$ for all regular graphs with $\mathcal{L}_k(N) > 1$ and $6 \leq N \leq 14$.

| $k \setminus N$ | 6 | 7 | 8 | 9 | 10 | 11 | 12 | 13 | 14 |
|---|---|---|---|---|---|---|---|---|---|
| 3 | 1 | 0 | 1 | 0 | 1 | 0 | 4 | 0 | 10 |
| 4 | 0 | 2 | 1 | 1 | 1 | 1 | 2 | 10 | 14 |
| 5 | 0 | 0 | 2 | 0 | 1 | 0 | 1 | 0 | 1 |
| 6 | 0 | 0 | 0 | 3 | 2 | 5 | 1 | 2 | 1 |
| 7 | 0 | 0 | 0 | 0 | 2 | 0 | 4 | 0 | 1 |
| 8 | 0 | 0 | 0 | 0 | 0 | 5 | 6 | 49 | 4 |
| 9 | 0 | 0 | 0 | 0 | 0 | 0 | 4 | 0 | 14 |
| 10 | 0 | 0 | 0 | 0 | 0 | 0 | 0 | 7 | 14 |
| 11 | 0 | 0 | 0 | 0 | 0 | 0 | 0 | 0 | 4 |

## 2.2. Relationships between Structure Coefficients and Graph Cycles

Recently, Giscard et al. [17] proposed an algorithm to count efficiently the number of cycles with length $\ell$ in a graph: $\mathcal{C}_\ell(N, k)$ with $3 \leq \ell \leq N$. Thus, it is feasible to count $\mathcal{C}_\ell(N, k)$ for all $\mathcal{L}_k(N)$ regular graphs with $N \leq 14$, as given in Table A1. As an example see Figure A1 with the count $\mathcal{C}_\ell(6, 3)$, $\ell = \{3, 4, 5, 6\}$, for the $\mathcal{L}_3(6) = 2$ graphs with $N = 6$ and $k = 3$. The following discussion is based on taking into account these numerical results.

In the previous section, it was shown that the maximal structure coefficients vary over interaction networks modeled as regular graphs, even if the number of players, coplayers, and cooperators is constant. Thus, it appears reasonable to assume that some features of the graphs may be associated with these differences. In the following, results are presented in support of an approximately linear relationship between the number of graph cycles with a certain length and the maximal structure coefficients. Two previous results can be interpreted as to point at the validity of such a relationship between the number of graph cycles and fixation properties. A first is from evolutionary games on lattice grids [18–21]. For these games, it has been shown that clusters of cooperators have a higher fixation probability than cooperators that are widely distributed on the grid. The location of the cluster on the grid does not matter. As lattice grids can be described by regular graphs (a Von Neumann neighborhood is a 4-regular graph and a Moore neighborhood is a 8-regular graph) clusters imply short and closed paths between the nodes of the grid. Furthermore, the grid means an abundance of cycles with even cycle length. A second result is that between the structure coefficients and the path length between the cooperators there is a strong negative correlation [14]. Cooperator path length is defined as the path length averaged over all pairs of cooperators on the evolutionary graph. If there are more than two cooperators, the cooperator path length has particularly small values if the cooperators

cluster next to each other and are linked by loops. Thus, small values of the cooperator path length correspond with the abundance of cycles of a certain length.

As there are $\mathcal{L}_k(N)$ regular graphs for a given $N$ and $k$, we obtain $\mathcal{L}_k(N)$ maximal structure coefficients $\sigma_{max_i}$, $i = 1, 2, \ldots \mathcal{L}_k(N)$ together with the same count of cycle length $\mathcal{C}_{\ell_i}(N, k)$. Thus, we may assume for each $i$ a linear relationship $\sigma_{max_i} = \mathcal{C}_{\ell_i}(N, k)x + \epsilon_i$ for some variables $x$ with an error term $\epsilon_i$. To test the validity of this linear relationship, we calculate the residual error

$$\text{res} = \frac{1}{\mathcal{L}_k(N)} \| \mathcal{C}_\ell x^* - \sigma_{max} \|, \tag{3}$$

where $\mathcal{C}_\ell$ comprises all $\mathcal{L}_k(N)$ cycle length $\mathcal{C}_{\ell_i}(N, k)$ and $\sigma_{max}$ contains all $\mathcal{L}_k(N)$ structure coefficients $\sigma_{max_i}$ for a given $N$ and $k$. The variable $x^*$ is the solution of the non-negative least square problem

$$x^* = \arg\min_x \| \mathcal{C}_\ell x - \sigma_{max} \|. \tag{4}$$

As the length of $x^*$ varies with varying $\mathcal{L}_k(N)$, the residual error in (3) is weighted by $\mathcal{L}_k(N)$ to make it comparable over all $N$ and $k$. Note that the residual error (3) gives equivalent results to the root–mean–square deviation, which is also sometimes used to measure the accuracy of a (linear) model. The results are given in Figure 4. We see that the residual error res is small for all $6 \leq N \leq 14$, $3 \leq k \leq N - 3$ and gets even smaller for $N$ getting larger. Generally, the error res is slightly larger for $k = 3$ and $k = N - 3$ than for intermediate values of $k$. This is also true for calculating res for each number of cooperators $c(\pi)$, see Figure 4b,c, which shows the results for $N = 12$ and $N = 14$. For $N = 14$ the values of res are generally smaller than for $N = 12$ and the largest values of res are obtained for small and large $k$ for all $c(\pi)$. To conclude, we can observe that the results for the residual error res are generally very small, which is equivalent to saying that the error term $\epsilon_i$ in the assumed linear relationship $\sigma_{max_i} = \mathcal{C}_{\ell_i}(N, k)x + \epsilon_i$ has an expected value $\mathbb{E}(\epsilon_i) \approx 0$. Thus, there is some justification to observe that between the maximal structure coefficients $\sigma_{max_i}$ and the cycle count $\mathcal{C}_{\ell_i}(N, k)$ there is an approximately linear relationship.

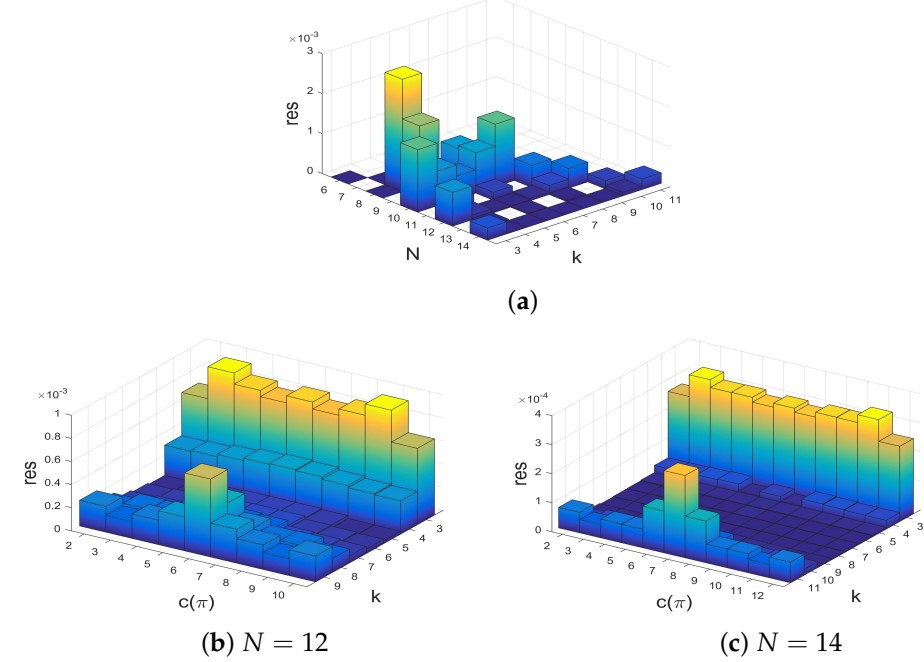

**(a)**

**(b)** $N = 12$      **(c)** $N = 14$

**Figure 4.** Residual error res according to Equation (3) over $N$, $k$, and $c(\pi)$: (**a**) over $N$ and $k$, (**b**) over $k$ and $c(\pi)$ for $N = 12$, and (**c**) over $k$ and $c(\pi)$ for $N = 14$.

Finally, another aspect of the interplay between the graph structure and fixation properties should be highlighted. To begin with, we analyze the cycle count $\mathcal{C}_\ell(N,k)$ of $\sigma_{max}$-graphs, which are those graphs among the $\mathcal{L}_k(N)$ regular graphs that have maximal structure coefficients. Consider the example $N = 12$ and $k = 3$. There are $\mathcal{L}_3(12) = 85$ graphs of which $\#_{\sigma_{max}} = 4$ are $\sigma_{max}$-graphs, compare Table 1 with Table A1. For these 4 graphs, we analyze how the count $\mathcal{C}_\ell(12,3)$ is distributed over $\ell = 3, 4, \ldots, 12$. A possible way to visualize such an analysis is based on schemaballs [13,22], see Figure 5a. In such a schemaball, we draw Bezier curves connecting the count $\mathcal{C}_\ell(N,k)$ in the upper half of the ball with the associated cycle length $\ell$ in the lower half. The actual values of both $\ell$ and $\mathcal{C}_\ell(N,k)$ are written on the ball. The curves are colored in such a way that equal values of the cycle length $\ell$ have the same (and specific) color, no matter to which cycle count $\mathcal{C}_\ell(N,k)$ they are belonging. The colors are selected equidistant from an RGB color wheel. If there are several $\sigma_{max}$-graphs, as there are $\#_{\sigma_{max}} = 4$ for $N = 12$, $k = 3$ in Figure 5a, each graph has its own set of curves between $\ell$ and $\mathcal{C}_\ell$. The schemaball thus contains all of them, which means there may be curves between the same value of $\ell$ and several $\mathcal{C}_\ell$ (and vice versa). For instance, in Figure 5a showing the schemaball for $N = 12$ and $k = 3$, we see that for $\ell = 3$, which is cycles of length 3, also known as triangles, we find connections to $\mathcal{C}_3(13,3) = (2,3,4,5)$. This means each of the $\#_{\sigma_{max}} = 4$ graphs has triangles, one has 2 of them, another one has 3, still another one has 4 and the last one has 5 triangles.

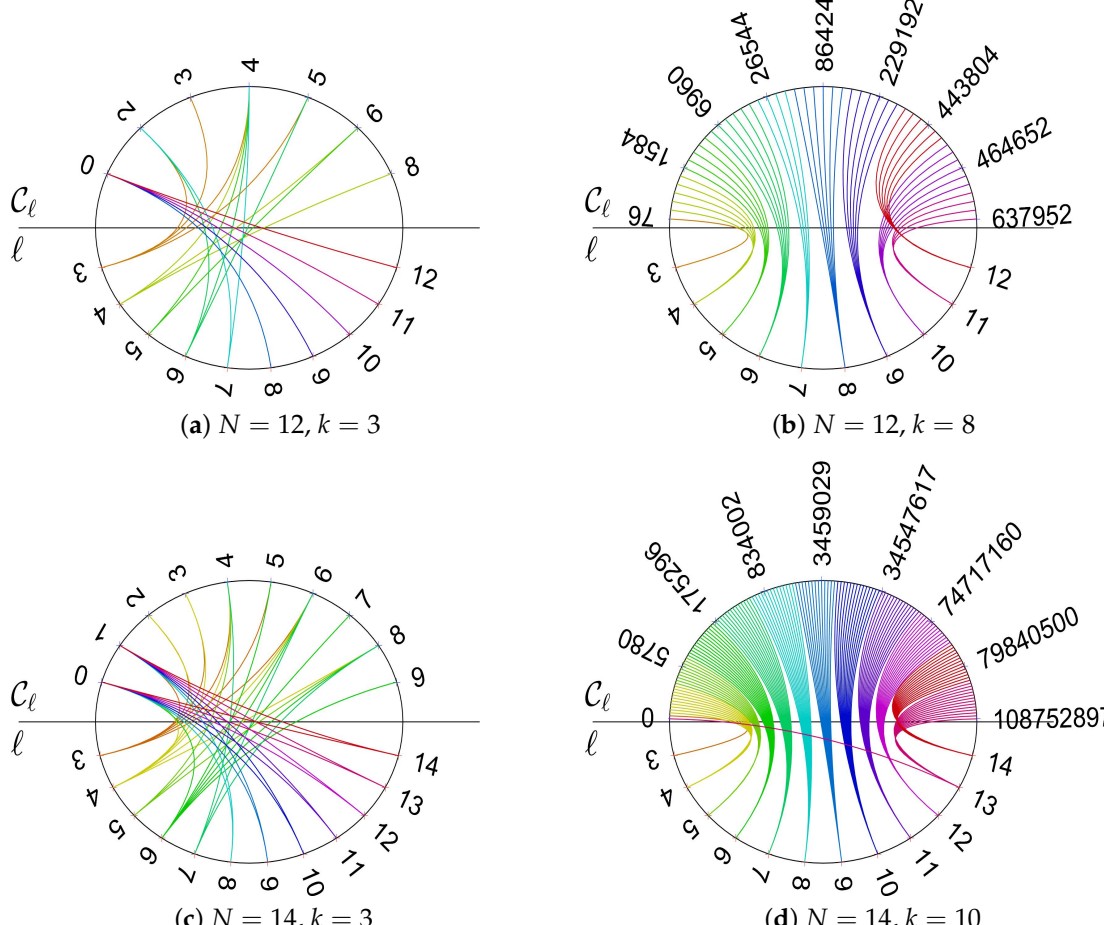

**(a)** $N = 12$, $k = 3$

**(b)** $N = 12$, $k = 8$

**(c)** $N = 14$, $k = 3$

**(d)** $N = 14$, $k = 10$

**Figure 5.** Examples of schemaballs of $\sigma_{max}$-graphs.

From the visualization using a schemaball it can be immediately seen that for $N = 12$ and $k = 3$ small cycles lengths, that is $\ell = \{3, 4, \ldots, 7\}$, have generally a count $\mathcal{C}_\ell(12,3) > 0$. For large cycle lengths, that is $\ell = \{8, 9, \ldots, 12\}$, we have $\mathcal{C}_\ell(12,3) = 0$. For $N = 14$ and $k = 3$, see Figure 5c, we get

very similar results. By contrast, for larger $k$, not only the cycle count $\mathcal{C}_\ell(N, k)$ is much higher than for lower $k$, but also the distribution over cycles lengths $\ell$ is quite different, see the examples $N = 12$, $k = 8$, Figure 5b and $N = 14$, $k = 10$, Figure 5d. Here, small as well as large cycle lengths $\ell$ have a substantial count $\mathcal{C}_\ell(N, k)$. Moreover, every cycle length $\ell$ is connected to a distinct interval of $\mathcal{C}_\ell(N, k)$. This means that the $\sigma_{max}$-graphs have very similar counts $\mathcal{C}_\ell(N, k)$ for each $\ell$. These properties become even more clear if we additionally consider the schemaballs for $\sigma_{min}$-graphs, which are the graphs with minimal structure coefficients see Figure 6 for the same examples as Figure 5. Not only there are more $\sigma_{min}$-graphs than $\sigma_{max}$-graphs, (for instance 77 vs. 4 for $N = 12$, $k = 3$, or 359 vs. 6 for $N = 12$, $k = 8$), the balls for small $k$ look very different, compare Figure 6a,c with Figure 5a,c. For the $\sigma_{min}$-graphs and small $k$ even large cycle length $\ell$ have a substantial count $\mathcal{C}_\ell(N, k)$. The count is actually much higher, which means that $\sigma_{min}$-graphs have generally more cycles of a given length than $\sigma_{max}$-graphs. On the other hand, for large $k$ the differences are rather marginal. The only difference is that the schemaballs are more dense, which means that $\sigma_{min}$-graphs have more different counts for a given cycle length than $\sigma_{max}$-graphs. For the other tested numbers of players, $N \leq 14$ similar results are obtained as shown in Figures 5 and 6. We next discuss some implications of these results for the evolution of cooperation on regular evolutionary graphs.

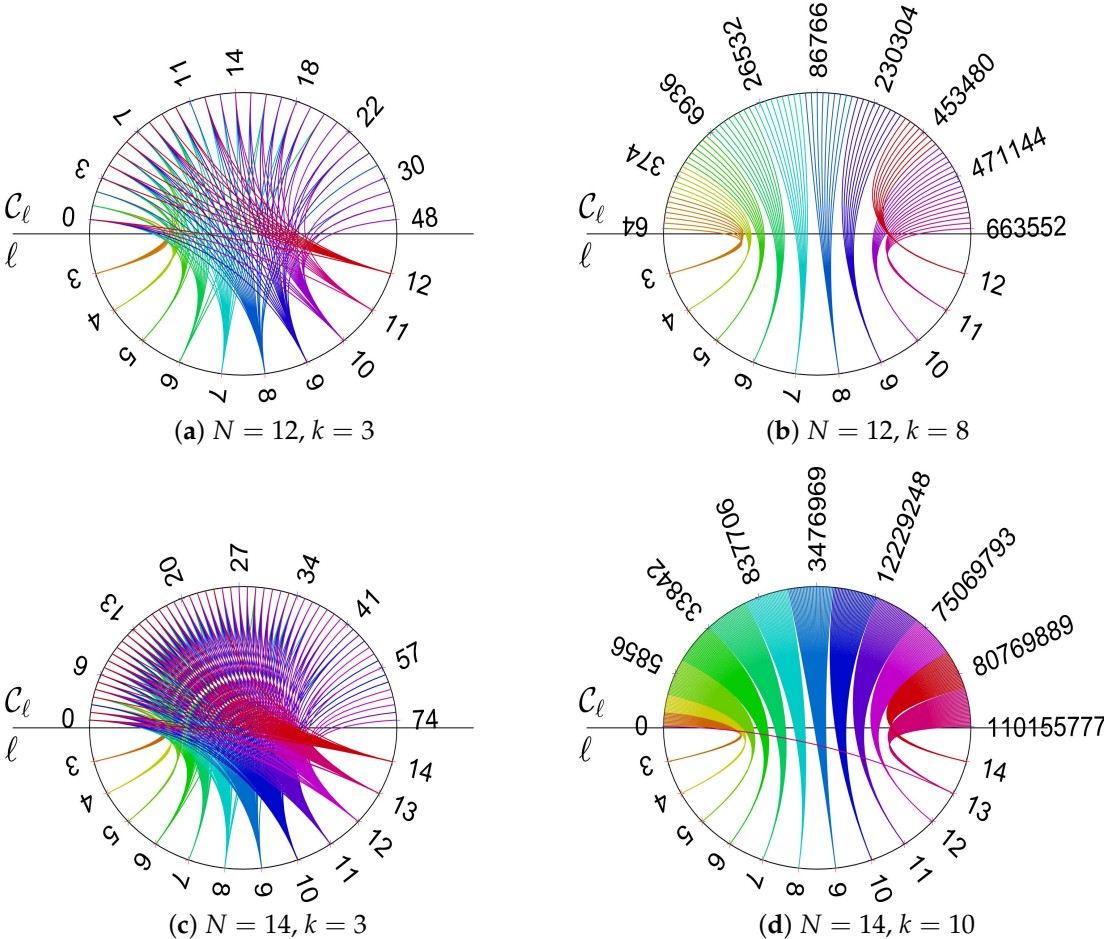

**Figure 6.** Examples of schemaball of $\sigma_{min}$-graphs.

## 3. Discussion and Conclusions

In this paper, structure coefficients $\sigma(\pi, \mathcal{G})$ introduced by Chen et al. [12] (see [13,14] for further analysis) are studied for DB updating and all regular interaction graphs with $N \leq 14$ players and $3 \leq k \leq N - 3$ coplayers. These structure coefficients provide a simple condition connecting long-term

prevalence of cooperation with the values of the payoff matrix (A1), the structure of the evolutionary graph $\mathcal{G}$, and the arrangement of any number of cooperators and defectors on this graph, which is expressed by the configuration $\pi$. Cooperation is favored for weak selection and a configuration $\pi$ on a graph $\mathcal{G}$ if

$$\sigma(\pi, \mathcal{G}) > \frac{c - b}{a - d}. \tag{5}$$

For $\sigma(\pi, \mathcal{G}) < 1$, the game favors the evolution of spite, which can be seen as a sharp opposite to cooperation. For $\sigma(\pi, \mathcal{G}) = 1$, the condition (5) matches the standard condition of risk–dominance. For $\sigma(\pi, \mathcal{G}) > 1$, the diagonal elements of the payoff matrix (A1), $a$ and $d$, are more critical than the off-diagonal elements, $b$ and $c$, for determining which strategy is favored. For instance, cooperation can be favored in the Prisoner's Dilemma game, which is specified by $c > a > d > b$. The condition (5) implies that a larger value of $\sigma(\pi, \mathcal{G})$ still allows cooperation to emerge if $a - d$ is small (or $c - b$ is large). For the Stag Hunt game (Coordination game), characterized by $a > c \geq d > b$, the condition $\sigma(\pi, \mathcal{G}) > 1$ means to favor a Pareto–efficient strategy ($a > d$) over a risk–dominant strategy ($a + b < c + d$). Again, a larger value of $\sigma(\pi, \mathcal{G})$ tolerates a smaller Pareto–efficiency $a - d$. Put differently, cooperation is favored even if the difference between reward and punishment is rather low. A generalization of these discussions can be achieved by the universal scaling approach for payoff matrices that facilitates studying a continuum of social dilemmas [23]. According to this approach, a larger value of $\sigma(\pi, \mathcal{G})$ implies a larger section of the parameter space spanned by gamble-intending and risk-averting dilemma strength [24]. Based on this interpretation of the structure coefficient $\sigma(\pi, \mathcal{G})$, we may review the following major results of the computational experiments presented in Section 2 and draw conclusions.

a.  There is an approximately linear relationship between maximal structure coefficients and the count of cycles of the interaction graph with a certain length. Moreover, the number of $\sigma_{max}$-graphs grows much slower for a rising number of players than the number of $k$-regular graphs on $N$ vertices. Thus, graphs with maximal structure coefficients get rare for the number of players $N$ getting large.

b.  The values of the structure coefficients are larger for a small number of coplayers (that is for graphs with a small degree) than for larger numbers of coplayers. It is maximal for $k = 3$, which is cubic graphs. This is also the case for the largest difference between maximal and minimal structure coefficients. Thus, for regular evolutionary graphs describing the interactions between players, the results for $N \leq 14$ players suggest that a smaller number of coplayers is particularly prone to promote cooperation if a favorable graph is selected. The selection of graphs does matter less for a larger number of coplayers. The $\sigma_{max}$-graphs with small numbers of coplayers $k$ not only have the largest maximal structure coefficients, they are also characterized by the absence of cycles with a length above a certain limit, see examples in the collection of $\sigma_{max}$-graphs in Appendix C.

c.  There are not only no long cycles in $\sigma_{max}$-graphs with small $k$. The graphs are also structured into blocks that are connected by cut vertices and/or hinge vertices. A cut vertex is a vertex whose removal disconnects the graph, while a hinge vertex is a vertex whose removal makes the distance longer between at least two other vertices of the graphs [25,26]. For instance, for $N = 12$ and $k = 3$, the vertices occupied by the players $\mathcal{I}_3$ and $\mathcal{I}_9$, see Figure A5, are cut vertices, while for $N = 10$ and $k = 4$, see Figure A4b, the vertices occupied by the players $\mathcal{I}_5$ and $\mathcal{I}_6$ are hinge vertices as their removal would make the distance between $\mathcal{I}_4$ and $\mathcal{I}_7$ longer. The blocks are occupied by clusters of cooperators. The clusters can be seen to serve as a mutant family that invades the remaining graph. As vertices with players of opposing strategies are connected by cut and/or hinge vertices there is only a small number of migration paths (or even just a single path) for the cooperators and/or defectors. A similar observation has been reported for evolutionary games on lattice grids [18,20], see also the discussion in Section 2.2. To summarize: the results suggest that $\sigma_{max}$-graphs for small numbers of coplayers have some distinct graph–theoretical

properties. Searching for these properties in a given graph may inform the design of interactions graphs that are either particularly prone to cooperation or particularly opposed to it.

d.　The property of missing long cycles is also a possible explanation as to why regular graphs with a small degree differ substantially from graphs with a larger degree in terms of promoting cooperation in evolutionary games. A larger degree makes it impossible to have blocks that are connected by only a few edges. As the number of edges increases linearly with the degree by $kN/2$ and each vertex has the same number of edges, there is an ample supply on connections. These results imply that the connectivity properties of the interaction graph play an important role in the emergence of cooperation. It may be interesting to see if these connectivity issues may possibly also show in algebraic graph measures, for instance, algebraic connectivity expressed by the Fiedler vector.

e.　The paper discusses the evolution of cooperation for all regular graphs with $N \leq 14$ players with death–birth (DB) updating and weak selection. Thus, it may be interesting to hypothesize about the relevance of the results for non-regular graphs, stronger levels of selection and other types of strategy updating. Recently, the relationships between the graph structure and fixation properties have been clarified substantially for a single mutant [6–9]. These results suggest that regular graphs have some similarities to Erdös–Rényi graphs, whereas other types of graphs, for instance, cycles, trees, stars or comet-kites are much more different. Thus, it might be possible that the results given in this paper for regular graphs may similarly apply to Erdös–Rényi graphs. Particularly interesting in this context may be the relationships between the connectivity of the graphs and the promotion of cooperation. Furthermore, it has been shown that extrapolating results from weak to intermediate and strong selection is not always possible and depends highly on game characteristics, population size and spatial heterogeneity of the network [5,27,28]. However, a comparison between fixation probabilities is rather robust for varying intensity of selection and a single cooperator [27]. Thus, the results obtained using structure coefficients may still be valid beyond weak selection. Finally, for weak selection, many fitness-based updating schemes and pairwise comparison with a Fermi function have similar fixation properties if the fitness can be approximated as a positive constant [29,30]. Thus, the results obtained for DB given in this paper might also have relevance to pairwise comparisons. On the other hand, it has also been shown that for an increasing level of selection intensity fitness-based models and pairwise comparison models of evolutionary games are typically different [31]. These brief comments about the relevance of the results for non-regular graphs, stronger levels of selection and other types of strategy updating must be treated with due caution as they are informed by results for a single mutant. Thus, further work is needed to clarify these relationships and see if they are also valid for multiple mutants (or any arrangement of cooperators and defectors).

f.　The results are given in this paper show a clear dependency between the long-term prevalence of cooperation in evolutionary games on regular graphs and some of their graph–theoretical properties, which generally confirm previous findings on clusters of cooperators in games on lattice grids [18–21], on pairs of mutants on a circle graph ($k = 2$) [32], and on short cooperator path lengths on some selected regular graphs with $N = 12$ and $k = 3$, among them the Frucht, the Tietze and the Franklin graph [14]. However, apart from statements about the prevalence of cooperation, there are also other quantifiers of evolutionary dynamics that are highly relevant. In other words, some of the difficulty in the given approach for evaluating the emergence of cooperation in evolutionary games on graphs arises from structure coefficients merely treating a comparison of fixation probabilities. The condition indicates that the fixation probability of cooperation is higher than the fixation probability of defection. This, however, does not entail the values of these probabilities. However, structure coefficients can be calculated with polynomial time complexity [12], while computing fixation probabilities is generally intractable due to an exponential time complexity [33–35]. In other words, by using the approach involving structure coefficients, we exchange computational tractability for obtaining just a comparison of fixation probabilities instead of their exact values. Moreover, the difference in the descriptive power

of the structure coefficients as compared to the fixation probabilities is salient in another way. Most likely, there is a rather complex relationship between structure coefficients and fixation probability, which is illustrated by the example of a single cooperator for which the structure coefficient does precisely not imply unique values of the fixation probability of cooperation. For a single cooperator, we get a single value of the structure coefficient, but fixation probabilities vary over initial configurations as shown for the Frucht and for the Tietze graph [36].

The discussion shows that calculating fixation probabilities and fixation times for multiple mutant configurations is not only computationally expensive, but also has a huge number of possible setups, for instance, which one of the considerable number of graphs to analyze, or where to place cooperators on the evolutionary graph and how many. There are various experimental parameters to be taken into account, which might be why so far systematically conducted computational studies are sparse. In this sense, another contribution of this paper might be seen in pointing at interesting settings for computational experiments calculating fixation probabilities and fixation times. The results given in this paper show that among all the regular interaction graphs with $N \leq 14$ players and $3 \leq k \leq N - 3$ coplayers, there is a comparably small number of graphs (as given in Table 1) which favor cooperation more than others. It may be interesting to see if these graphs also stand out in terms of fixation probability and fixation time.

**Funding:** This research received no external funding.

**Acknowledgments:** I wish to thank Markus Meringer for making available the `genreg` software [37] used for generating the regular graphs according to Table A1 and for helpful discussions.

**Conflicts of Interest:** The author declares no conflict of interest.

## Abbreviations

The following abbreviations are used in this manuscript:

BD    Birth Death updating
DB    Death Birth updating

## Appendix A. Configurations, Regular Graphs, and Structure Coefficients

The coevolutionary games we consider here have $N$ players $\mathcal{I} = \{\mathcal{I}_i\}$, $i = 1, 2, \ldots, N$, that each uses either of two strategies $\pi_i \in \{C, D\}$, which we may interpret as cooperating or defecting. Each player $\mathcal{I}_i$, while interacting with a coplayer $\mathcal{I}_j$, receives a payoff according to the $2 \times 2$ payoff matrix

$$
\begin{array}{c}
i \backslash^j \\
C \\
D
\end{array}
\begin{array}{cc}
C & D \\
\begin{pmatrix} a & b \\ c & d \end{pmatrix}
\end{array}. \tag{A1}
$$

Which player interacts with whom is described by the interaction graph $\mathcal{G} = (V, E)$, where the vertices $v_i \in V$ represent the players and the edges $e_{ij} \in E$ indicate that the players $\mathcal{I}_i$ and $\mathcal{I}_j$ interact as mutual coplayers [11,38,39]. Which strategy is used by which player at a given point of time is specified by a configuration $\pi = (\pi_1, \pi_2, \ldots, \pi_N)$ with $\pi_i \in \{C, D\}$. If we represent the two strategies by a binary code $\{C, D\} \to \{1, 0\}$, a configuration appears as a binary string the Hamming weight of which denotes the number of cooperators $c(\pi)$. For games with $N$ players, there are $2^N$ configurations with 2 configurations ($\pi = (00\ldots0)$ and $\pi = (11\ldots1)$) absorbing. Players may update their strategies in an updating process, for instance death–birth (DB) or birth–death (BD) updating [40,41]. Recently, it was shown by Chen et al. [12] that strategy $\pi_i = 1 = C$ is favored over $\pi_i = 0 = D$ if

$$
\sigma(\pi, \mathcal{G})(a - d) > (c - b). \tag{A2}
$$

These results apply to weak selection and $2 \times 2$ games with $N$ players, payoff matrix (A1), any configuration $\pi$ of cooperators and defectors and any interaction network modeled by a simple, connected, $k$-regular graph. The quantity $\sigma(\pi, \mathcal{G})$ in Equation (A2) is the structure coefficient of the configuration $\pi$ and the graph $\mathcal{G}$. It may not have the same value for different arrangements of cooperators and defectors described by the configuration $\pi$ and the same graph $\mathcal{G}$, but also for the same configuration $\pi$ and different interaction networks modeled by a regular graph $\mathcal{G}$. In particular, it was shown that for weak selection and the graph $\mathcal{G}$ describing interaction as well as replacement graph, the structure coefficient $\sigma(\pi, \mathcal{G})$ can be calculated with time complexity $\mathcal{O}(k^2 N)$ for DB and BD updating [12]. For DB updating there is

$$\sigma(\pi, \mathcal{G}) = \frac{N\,(1 + 1/k)\,\overline{\omega^1} \cdot \overline{\omega^0} - 2\overline{\omega^{10}} - \overline{\omega^1 \omega^0}}{N\,(1 - 1/k)\,\overline{\omega^1} \cdot \overline{\omega^0} + \overline{\omega^1 \omega^0}}, \tag{A3}$$

with 4 local frequencies ($\overline{\omega^1}$, $\overline{\omega^0}$, $\overline{\omega^{10}}$ and $\overline{\omega^1 \omega^0}$), which depend on $\pi$ and $\mathcal{G}$, see [12–14] for a probabilistic interpretation of these frequencies. Our focus here is on DB updating as it has been shown that BD updating never favors cooperation [12].

## Appendix B. Isomorphic Graphs, Isomorphic Configurations, and Cycle Counts

The structure coefficient $\sigma(\pi, \mathcal{G})$, as for instance defined for DB updating by Equation (A3), may vary over configurations $\pi$ and graphs $\mathcal{G}$. This suggests the question of upper and lower bounds of $\sigma(\pi, \mathcal{G})$. For a rather low number of players, it appears feasible to check all $\sigma(\pi, \mathcal{G})$, as demonstrated in the paper for $N \leq 14$ and all regular graphs with up to 14 vertices. For a $2 \times 2$ game with $N$ players, there are $2^N - 2$ non-absorbing configurations $\pi$. These configurations can be grouped according to the number of cooperators $c(\pi)$, $2 \leq c(\pi) \leq N - 2$. The number of simple, connected regular graphs is known for small numbers of vertices, e.g., [37], see Table A1. Note that these numbers apply to graphs that are all not isomorphic with each isomorphism class being represented by exactly one graph. In other words, Table A1 also gives the number of isomorphism classes for all $6 \leq N \leq 14$ and $3 \leq k \leq N - 1$. Isomorphism refers to the property that two graphs are structurally alike and merely differ in how the vertices and edges are named. More precisely, two graphs are isomorphic if there is a bijective mapping $\theta$ between their vertices which preserves adjacency [42], pp. 12–14.

**Table A1.** The numbers $\mathcal{L}_k(N)$ of simple connected $k$-regular graphs on $N$ vertices see e.g., [37], which corresponds to the number of regular interaction graphs with $N$ players and $k$ coplayers for $6 \leq N \leq 14$ and $3 \leq k \leq N - 1$. Note that only for $k \leq N - 3$ there is more than one graph: $\mathcal{L}_k(N) > 1$.

| $k \setminus N$ | 6 | 7 | 8 | 9 | 10 | 11 | 12 | 13 | 14 |
|---|---|---|---|---|---|---|---|---|---|
| 3 | 2 | 0 | 5 | 0 | 19 | 0 | 85 | 0 | 509 |
| 4 | 1 | 2 | 6 | 16 | 59 | 265 | 1.544 | 10.778 | 88.168 |
| 5 | 1 | 0 | 3 | 0 | 60 | 0 | 7.848 | 0 | 3.459.383 |
| 6 | 0 | 1 | 1 | 4 | 21 | 266 | 7.849 | 367.860 | 21.609.300 |
| 7 | 0 | 0 | 1 | 0 | 5 | 0 | 1.547 | 0 | 21.609.301 |
| 8 | 0 | 0 | 0 | 1 | 1 | 6 | 94 | 10.786 | 3.459.386 |
| 9 | 0 | 0 | 0 | 0 | 1 | 0 | 9 | 0 | 88.193 |
| 10 | 0 | 0 | 0 | 0 | 0 | 1 | 1 | 10 | 540 |
| 11 | 0 | 0 | 0 | 0 | 0 | 0 | 1 | 0 | 13 |
| 12 | 0 | 0 | 0 | 0 | 0 | 0 | 0 | 1 | 1 |
| 13 | 0 | 0 | 0 | 0 | 0 | 0 | 0 | 0 | 1 |

Consider, for example, the $\mathcal{L}_3(6) = 2$ interaction graphs with $N = 6$ players, each with $k = 3$ coplayers, see Figure A1. For the graph in Figure A1a we get the maximal structure coefficient

$\sigma_{max} = 1.1818$ for 2 configurations, $\pi = (111000)$ as shown in Figure A1a and $\pi = (000111)$. By the isomorphism

$$\theta = \begin{pmatrix} v_1 & v_2 & v_3 & v_4 & v_5 & v_6 \\ 1 & 2 & 3 & 4 & 5 & 6 \\ 6 & 1 & 2 & 3 & 4 & 5 \end{pmatrix}$$

we obtain an isomorphic graph as shown in Figure A1b. For this graph, the configuration $\pi = (111000)$ has $\sigma = 1.0000$, but $\pi = (110001)$ and $\pi = (001110)$ have $\sigma_{max} = 1.1818$. Note that between the configurations with $\sigma_{max}$ the same isomorphic mapping $\theta$ applies. In other words, the structure coefficients are invariant under isomorphic mappings. For each pair of isomorphic graphs, there are isomorphic configurations that have the same value of the structure coefficient. For the graph in Figure A1c, we obtain the result that the structure coefficient is constant over all configurations (except the absorbing configurations). Thus, isomorphic transformations do not alter the values of $\sigma(\pi, \mathcal{G})$.

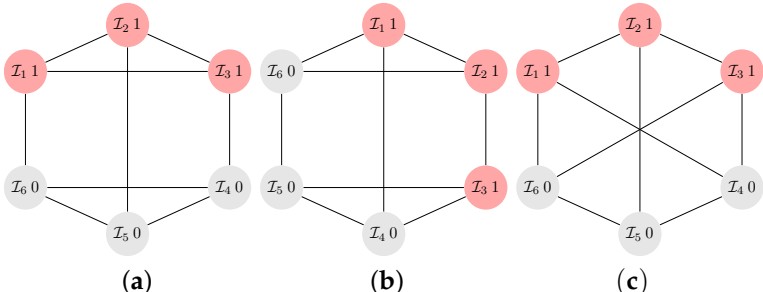

**Figure A1.** The $\mathcal{L}_3(6) = 2$ interaction graphs with $N = 6$ players, each with $k = 3$ coplayers. All are vertex-transitive and (**c**) is even symmetric (edge-transitive). The graph in (**a**) has a maximal structure coefficient $\sigma_{max} = 1.1818$, which is obtained for two configurations with $c(\pi) = 3$ cooperators: $\pi = (111000)$ (as shown in (**a**)) and $\pi = (000111)$. For the isomorphic graph in (**b**), we get $\sigma_{max} = 1.1818$ for the isomorphic configurations $\pi = (110001)$ and $\pi = (001110)$. The graph in (**c**) has the same structure coefficient $\sigma = 1.0000$ for all configurations. Regarding the count of cycles with length $\ell$, we see that the graphs in (**a**) and (**b**) have $\mathcal{C}_{\ell_1}(6,3) = (2,3,6,2)$, while for the graph in (**c**) there is $\mathcal{C}_{\ell_2}(6,3) = (0,9,0,6)$.

These results apply generally to structure coefficients $\sigma(\pi, \mathcal{G})$ of regular graphs. The local frequencies in Equation (A3) solely depend on counting two types of paths on the interaction graph [12–14]. The quantities $\overline{\omega^1}$, $\overline{\omega^0}$, and $\overline{\omega^1\omega^0}$ relate to the number of paths with length 1 that connect any vertex with adjacent vertices that hold a cooperator (or defector). The quantity $\overline{\omega^{10}}$ relates to the number of paths with length 2 from any vertex to adjacent vertices on which the first vertex of the path holds a cooperator and the second vertex holds a defector. As an isomorphic reshuffling of vertices preserves adjacency, these numbers stay the same if the isomorphism acts on both the vertices and the configurations. Thus, suppose two graphs $\mathcal{G}_i$ and $\mathcal{G}_j$ are isomorphic with isomorphism $\theta$. Then, it follows $\sigma(\pi, \mathcal{G}_i) = \sigma(\theta(\pi), \mathcal{G}_j)$. Consequently, the maximal structure coefficient is invariant as well, that is for isomorphic graphs $\mathcal{G}_i$ and $\mathcal{G}_j$ there is $\sigma_{max_i} = \max_{\pi} \sigma(\pi, \mathcal{G}_i) = \sigma_{max_j} = \max_{\pi} \sigma(\pi, \mathcal{G}_j)$. Any regular graph belongs to one of the isomorphism classes and can be obtained by isomorphic transformations by any member of this class. Regular interaction graphs that are isomorphic have the same distribution of structure coefficients $\sigma(\pi, \mathcal{G})$ over the number of cooperators $c(\pi)$. Thus, by considering one representative of each isomorphism class, we can make statements about structure coefficients for all regular graphs.

For each graph, there is a specific count $\mathcal{C}_\ell(N, k)$ of cycles with length $\ell$, $3 \leq \ell \leq N$. There are efficient algorithms to count these cycles [17]. Consider again the $\mathcal{L}_3(6) = 2$ graphs with $N = 6$ players and $k = 3$ coplayers, see Figure A1. We find the graph in Figure A1a,b has $\mathcal{C}_{\ell_1}(6,3) = (2,3,6,2)$ with $\ell = \{3,4,5,6\}$ (there are 2 cycles of length $\ell = 3$, 3 cycles of length $\ell = 4$, 6 cycles of length $\ell = 5$, 2 cycles of length $\ell = 6$), while the graph in Figure A1c has $\mathcal{C}_{\ell_2}(6,3) = (0,9,0,6)$. It generally applies that isomorphic graphs have the same $\mathcal{C}_\ell(N, k)$. Graphs that are not isomorphic have frequently a distinct

count $\mathcal{C}_\ell(N, k)$, but there are also cases, particularly for $N$ getting larger, where 2 not isomorphic graphs have the same count $\mathcal{C}_\ell(N, k)$.

**Appendix C. Collection of $\sigma_{max}$-Graphs with $N \leq 14$**

We here give a collection of selected $\sigma_{max}$-graphs with $N \leq 14$. The graphs are shown to illustrate some graph–theoretical properties associated with the prevalence of cooperation. The single $\sigma_{max}$-graph with $N = 6$ is already shown in Figure A1a. For $N = 7$, there are $\mathcal{L}_4(7) = 2$ regular graphs, which both have the same maximal structure coefficients. In other words, the count of graphs equals the count of $\sigma_{max}$-graph, which is why they are not included in the collection.

Figures A2–A4 show all $\sigma_{max}$-graphs for $N = 8, 9, 10$ and $3 \leq k \leq N - 3$ together with the values of $\sigma_{max}$ and the associated configurations. For $N = 12$ and $N = 14$, only some examples of $\sigma_{max}$-graphs are given in Figures A5–A7 due to brevity. A full list of all $\sigma_{max}$-graphs for $11 \leq N \leq 14$ and $3 \leq k \leq N - 3$ is made available here [43]. It is particularly noticeable that the $\sigma_{max}$-graphs are structured to have blocks with clusters of mutants. For instance, we see such a block with $(\mathcal{I}_1, \mathcal{I}_2, \mathcal{I}_3, \mathcal{I}_4)$ for the graph with $N = 8$ and $k = 3$ in Figure A2a and for $N = 9$ and $k = 4$ in Figure A3a, or for $N = 10$ and $k = 3$, Figure A4a and for the cubic graphs ($k = 3$) with $N = 12$ and $N = 14$ as well, see Figures A5 and A7. The $\sigma_{max}$-graphs with a larger degree ($=$ coplayers) still somewhat retain a "blockish" appearance (for instance $(\mathcal{I}_1, \mathcal{I}_2, \mathcal{I}_3, \mathcal{I}_4, \mathcal{I}_5)$ in Figure A4c) but to a far lesser degree. In addition, $\sigma_{max}$-graphs with larger degree are frequently vertex-transitive (for instance Figures A3d and A4e,g) which is not the case for cubic ($k = 3$) and quartic ($k = 4$) $\sigma_{max}$-graphs with $N \leq 14$, with the exception of $N = 6$ and $k = 3$, see Figure A1a. Furthermore, it can be observed that the blocks are occupied by clusters of cooperators, which are frequently connected by cut vertices and/or hinge vertices. For instance, for $N = 12$ and $k = 3$, the vertices occupied by the players $\mathcal{I}_3$ and $\mathcal{I}_9$, see Figure A5, are cut vertices, while for $N = 10$ and $k = 4$, see Figure A4b, the vertices occupied by the players $\mathcal{I}_5$ and $\mathcal{I}_6$ are hinge vertices as their removal would make the distance between $\mathcal{I}_4$ and $\mathcal{I}_7$ longer. As discussed above, the clusters can be seen to serve as a mutant family that invades the remaining graph. As vertices with players of opposing strategies are connected by cut and/or hinge vertices there is only a small number of migration paths (or even just a single path) for the cooperators and/or defectors.

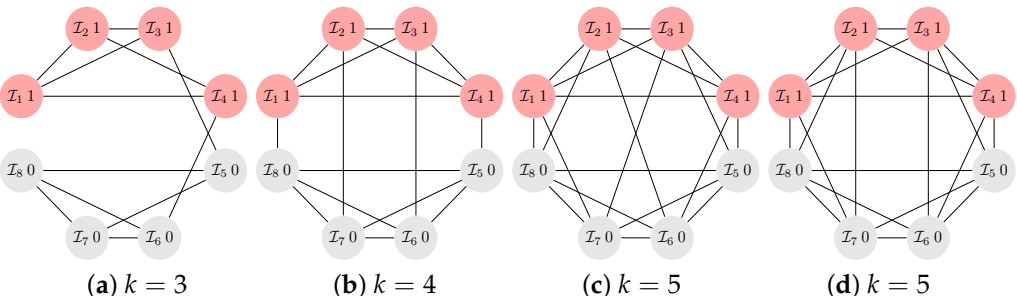

(a) $k = 3$  (b) $k = 4$  (c) $k = 5$  (d) $k = 5$

**Figure A2.** The $\sigma_{max}$-graphs for $N = 8$ and $k = 3, 4, 5$. We get $\sigma_{max} = 1.6538$ for $k = 3$, (**a**), $\sigma_{max} = 1.2222$ for $k = 2$, (**b**), and $\sigma_{max} = 0.9565$ for the 2 $\sigma_{max}$-graphs with $k = 5$, (**c,d**), each for the configuration $\pi = (1111\,0000)$. In addition, the same structure coefficient is obtained also for the configuration $\pi = (0000\,1111)$, and only for (**d**) additionally for $\pi = (1100\,0011)$ and $\pi = (0011\,1100)$.

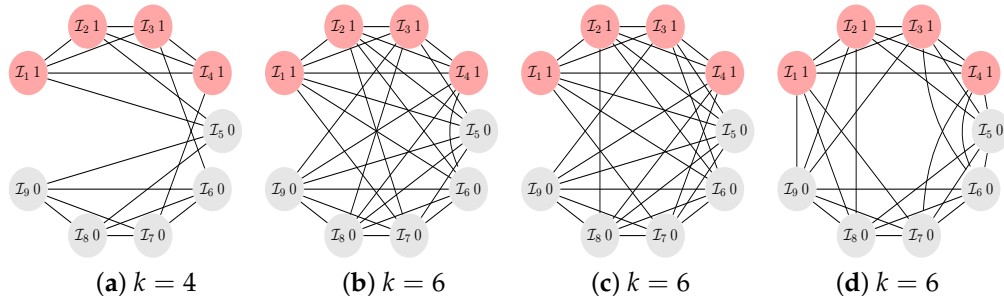

**Figure A3.** The $\sigma_{max}$-graphs for $N = 9$ and $k = 4$, 6. We get $\sigma_{max} = 1.3206$ for $k = 4$ (**a**) and the configuration $\pi = (11110\,0000)$, but also for $\pi = (11111\,0000)$, $\pi = (00000\,1111)$ and $\pi = (00001\,1111)$. For $k = 6$, there are 3 $\sigma_{max}$-graphs, (**b–d**), each with $\sigma_{max} = 0.9115$ for the configuration $\pi = (11110\,0000)$. There are several more configurations that have the same $\sigma_{max}$ due to the symmetry properties of these 3 graphs.

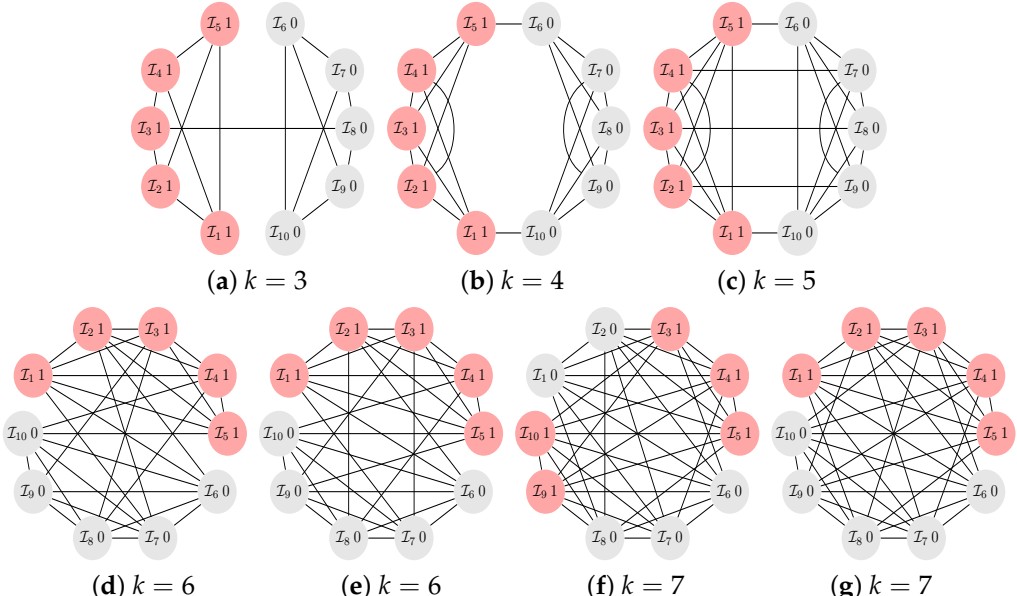

**Figure A4.** The $\sigma_{max}$-graphs for $N = 10$ and $k = 3, 4, \ldots, 7$. We get $\sigma_{max} = 1.8831$ for $k = 3$ (**a**), $\sigma_{max} = 1.5128$ for $k = 4$ (**b**), $\sigma_{max} = 1.2222$ for $k = 5$ (**c**), $\sigma_{max} = 1.0241$ for $k = 6$ (**d,e**) and $\sigma_{max} = 0.9145$ for $k = 7$ (**f,g**), all for the configuration $\pi = (11111\,00000)$, and also for $\pi = (00000\,11111)$. For one graph with $k = 7$, (**f**) the maximal structure coefficient is also obtained for 2 more configurations.

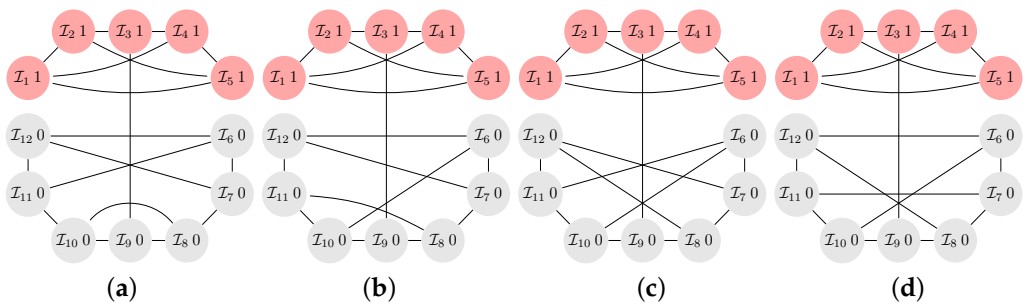

**Figure A5.** The 4 different $\sigma_{max}$-graphs (**a–d**) for $N = 12$ and $k = 3$, each with $\sigma_{max} = 1.9159$ for the configuration $\pi = (1111\,1000\,0000)$ (and also for $\pi = (0000\,0111\,1111)$).

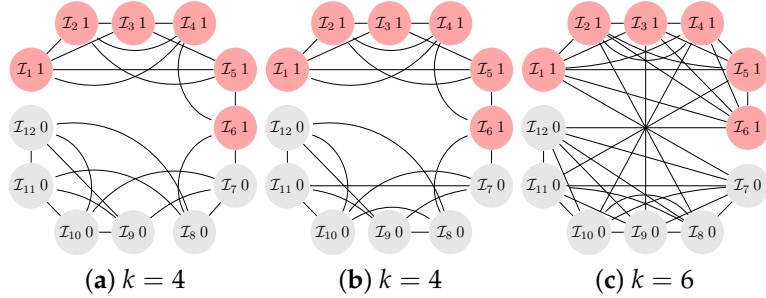

**Figure A6.** The $\sigma_{max}$-graphs for $N = 12$ and $k = 4$, 6. We have $\sigma_{max} = 1.5701$ for $k = 4$ (**a**,**b**) and $\sigma_{max} = 1.2105$ for $k = 6$ (**c**), each for the configuration $\pi = (1111\,1100\,0000)$ (and also for $\pi = (0000\,0011\,1111)$).

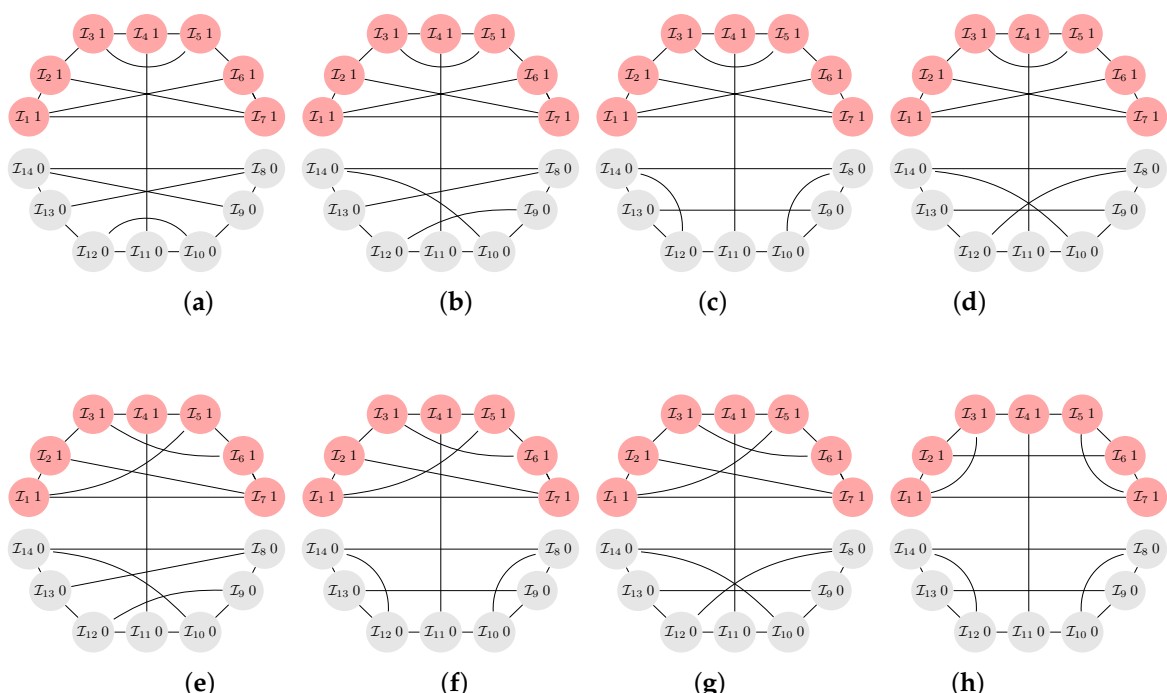

**Figure A7.** The $\sigma_{max}$-graphs for $N = 14$ and $k = 3$, each with $\sigma_{max} = 1.9396$ for the configuration $\pi = (1111\,1110\,0000\,00)$ (and also for $\pi = (0000\,0001\,1111\,11)$). Only 8 out of $\#_{sigma_{max}} = 10$ $\sigma_{max}$-graphs according to Table 1 are depicted. The remaining 2 graphs arise from the blocks depicted in the figures. If we call the upper half of the graphs in (**a**–**d**) the A–block, then the lower half of these graphs consists of blocks A, B, C, and D. The blocks are joined by the edge connecting $\mathcal{I}_4$ and $\mathcal{I}_{11}$. The graphs in (**e**–**h**) also are block-like with blocks B–B, B–C, B–D, and C–C. The 2 remaining graphs (not depicted) are formed by connecting the blocks C–D and D–D.

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
