# Peer review of "Evolution of Cooperation for Multiple Mutant Configurations on All Regular Graphs with N ≤ 14 Players"

_games, doi:10.3390/g11010012_

Round 1
Reviewer 1 Report
This paper study the emergence of cooperation in structured populations with arbitrary arrangements of cooperators and defectors on the graph defining the population structure. The author or authors use structure coefficients defined for configurations describing arrangements of any number of mutants to obtain results for weak selection to favor cooperation over defection on any regular graph with N ≤ 14 players.
The paper is a good quality technical contribution to the area and represents a useful addition to the literature. The paper is well written and logically structured, with the main results being discussed in the body of the paper while the derivations are collected in the appendices. I have only a few suggestions for further points that the authors might like to consider.
(1) The authors study birth-death updating in detail, but given the correspondence between birth-death updating and pairwise comparison Fermi updating considered in Ohtsuki & Nowak (2006), presumably similar results hold also for Fermi updating - perhaps the author or authors could briefly discuss this case.
(2) How fundamental in the assumption of regularity? Is it possible to obtain similar results on graphs that are "close" to regular in some suitable sense? For example, is it possible to obtain similar results for Erdős–Rényi graphs. Also is there any reason to believe that graphs with large variance in their degree distribution (such as power-law graphs) will have different evolutionary behavior?
(3) Can the authors offer any results or speculation going beyond weak selection, for example, with an exponential payoff to fitness mapping.
(4) The references are comprehensive, but an additional references which might be relevant in a general sense is Iyer & Killingback (2016) Evolution of Cooperation in Social Dilemmas on Complex Networks. PLoS Comput Biol 12(2): e1004779. doi:10.1371/journal.pcbi.1004779.
Author Response
(1) The authors study birth-death updating in detail, but given the correspondence between birth-death updating and pairwise comparison Fermi updating considered in Ohtsuki & Nowak (2006), presumably similar results hold also for Fermi updating - perhaps the author or authors could briefly discuss this case.
Response: For weak selection many fitness-based updating schemes and pairwise comparison with a Fermi function have similar fixation properties if the fitness can be approximated as a positive constant. Thus, the results obtained for BD given in this paper might also have relevance to pairwise comparison. On the other hand, it has also been shown that for an increasing level of selection intensity fitness-based models and pairwise comparison models of evolutionary games are typically different. A comment has been added in 3. Discussion and Conclusions.
(2) How fundamental in the assumption of regularity? Is it possible to obtain similar results on graphs that are "close" to regular in some suitable sense? For example, is it possible to obtain similar results for Erdős–Rényi graphs. Also is there any reason to believe that graphs with large variance in their degree distribution (such as power-law graphs) will have different evolutionary behavior?
Response: Recently, the relationships between the graph structure and fixation properties have been clarified substantially for a single mutant. These results indeed suggest that regular graphs have some similarity to Erd{\"o}s--Rényi graphs, whereas other types of graphs, for instance cycles, trees, stars or comet--kites are much more different. Thus, it might be possible that the results given in this paper for regular graphs may similarly apply to Erd{\"o}s--Rényi graphs. A comment has been added in 3. Discussion and Conclusions.
(3) Can the authors offer any results or speculation going beyond weak selection, for example, with an exponential payoff to fitness mapping.
Response: It was shown in the Literature that extrapolating results from weak to intermediate and strong selection is not always possible and depends highly on game characteristics, population size and spatial heterogeneity of the network. However, a comparison between fixation probabilities is rather robust for a varying intensity of selection and a single cooperator. Thus, the results obtained using structure coefficients may apply beyond weak selection. A comment has been added in 3. Discussion and Conclusions.
(4) The references are comprehensive, but an additional references which might be relevant in a general sense is Iyer & Killingback (2016) Evolution of Cooperation in Social Dilemmas on Complex Networks. PLoS Comput Biol 12(2): e1004779. doi:10.1371/journal.pcbi.1004779.
Response: I have added the references.
Reviewer 2 Report
This paper investigates the propensity for death-birth updating to select cooperative behavior on evolutionary interaction graphs. Specifically, it presents computations of the structural coefficient described by Chen et al. (2016) for regular interaction graphs with less than 15 players. This is a useful contribution to the literature on evolutionary graph theory as Chen et al. (2016) has shown that cooperation is favored under death-birth updating if the structural coefficient is sufficiently large and the intensity of selection is sufficiently weak.
In particular, it would be helpful if the nature of the papers main contribution was more explicitly stated in the abstract and the introduction. The current version of the abstract and introduction does not make it sufficiently clear that the main results are computational rather than analytical. The authors find relationships between the maximal structure coefficients and other features of the interaction graph such as the degree of the graph, the number of players, and the number of cycles with a certain length. These relationships to have been observed via computation of the structural coefficients. This should be stated more explicitly.
Author Response
Point: In particular, it would be helpful if the nature of the papers main contribution was more explicitly stated in the abstract and the introduction. The current version of the abstract and introduction does not make it sufficiently clear that the main results are computational rather than analytical. The authors find relationships between the maximal structure coefficients and other features of the interaction graph such as the degree of the graph, the number of players, and the number of cycles with a certain length. These relationships to have been observed via computation of the structural coefficients. This should be stated more explicitly.
Response: Thank you for your remark. I have updated the Abstract and the Introduction and stated that the approach and the results are computational.